# A Review of Safety Risk Theories and Models and the Development of a Digital Highway Construction Safety Risk Model

Loretta Bortey [1], David J. Edwards [1,2,*] , Chris Roberts [1] and Iain Rillie [3]

[1]  Department of the Built Environment, Birmingham City University, City Centre Campus, Millennium Point, Birmingham B4 7XG, UK; loretta.bortey@mail.bcu.ac.uk (L.B.); chri51988@live.com (C.R.)
[2]  Faculty of Engineering and the Built Environment, University of Johannesburg, Johannesburg 2092, South Africa
[3]  Health and Safety Technical Innovation Specialist, Safety, Engineering and Standards, National Highways Company Limited, Guildford GU1 4LZ, UK; iain.rillie@highwaysengland.co.uk
*   Correspondence: drdavidedwards@aol.com

**Abstract:** This study conducts a systematic review of safety risk models and theories by summarizing and comparing them to identify the best strategies that can be adopted in a digital 'conceptual' safety risk model for highway workers' safety. A mixed philosophical paradigm was adopted (that used both interpretivism and post-positivism couched within inductive reasoning) for a systematic review and comparative analysis of existing risk models and theories. The underlying research question formulated was: can existing models and theories of safety risk be used to develop this proposed digital risk model? In total, 607 papers (where each constituted a unit of analysis and secondary data source) were retrieved from Scopus and analysed through colour coding, classification and scientometric analysis using VOSViewer and Microsoft Excel software. The reviewed models were built on earlier safety risk models with minor upgrades. However, human elements (human errors, human risky behaviour and untrained staff) remained a constant characteristic, which contributed to safety risk occurrences in current and future trends of safety risk. Therefore, more proactive indicators such as risk perception, safety climate, and safety culture have been included in contemporary safety risk models and theories to address the human contribution to safety risk events. Highway construction safety risk literature is scant, and consequently, comprehensive risk prevention models have not been well examined in this area. Premised upon a rich synthesis of secondary data, a conceptual model was recommended, which proposes infusing machine learning predictive models (augmented with inherent resilient capabilities) to enable models to adapt and recover in an event of inevitable predicted risk incident (referred to as the resilient predictive model). This paper presents a novel resilient predictive safety risk conceptual model that employs machine learning algorithms to enhance the prevention of safety risk in the highway construction industry. Such a digital model contains adaptability and recovery mechanisms to adjust and bounce back when predicted safety risks are unavoidable. This will help prevent unfortunate events in time and control the impact of predicted safety risks that cannot be prevented.

**Keywords:** safety risk; models/theories; machine learning; prediction; resilience

## 1. Introduction

UK construction workers are five times more likely to be killed at work than other industries combined [1]. According to the Health and Safety Executive [2], construction workers' fatality rate is three times that of all other industries, even though the sector accounts for only 7% of the national workforce. Indeed, globally, the construction industry is noted for having one of the worst records in occupational health and safety [3,4]. This indicates that despite notable improvements in safety since the introduction of legislative

instruments (such as the UK's Occupational Safety and Health Act of 2004), accidents and injury continue unabated [5]. Perhaps one major reason is that construction personnel work in a dynamic and continuously changing environment, where new hazards and risks are sometimes not identified in the early planning stages.

Because of this perpetual state of flux, safety risk theories and models have been relied upon to create a safe working environment [6]. Hillson [7] has defined risk holistically as an uncertain event or condition that, if it occurs, has a positive or negative effect on a project's objectives. This definition, however, does not explicitly include workers or their safety. Consequently, the HSE defines 'safety risk' as: "the likelihood that a person may be harmed or suffers adverse health effects if exposed to a hazard" [2]. This definition considers the existence of hazards and the probability that a person may inadvertently interact with these hazards and become injured. Risk management, therefore, provides a system to control these risks [8]. Risk mitigation is critical during the project's design and planning phases of development, as many risks can be 'designed out', albeit not totally eliminated [9,10].

Techniques used to gain insights and minimize accidents and injuries in construction have relied on various models and theories for developing appropriate risk mitigation strategies. These have been applied in different contexts, ranging from occupational health [11] to the safety behaviours of workers [12] to constructing emergency safe refuges [13]. A review of diverse health and safety risk models and theories reveals theory and model application in several high-risk industries, such as mining, aviation, oil and gas, and medicine [11,14]. However, considering the construction industry, scant research has been undertaken on the effective application of these theories and models in a highway setting, thus resulting in limited safety research in this field. The various types of safety risk models and theories adopted in other fields could be applied to highway construction safety strategies by leveraging their most useful characteristics [13,15]. Hence, this study reviews these safety risk models and theories by summarising and comparing them to identify the best strategies that can be adopted in a digital 'conceptual' safety risk model. Associated objectives are to critically assess and understand the prevailing academic discourse on constructing theories and models of safety risk; classify existing safety risk models and theories and compare based on their characteristics; identify any knowledge gaps in the safety risk theories and models previously presented; and develop a novel digital safety risk conceptual model for highway construction safety. Such a model provides the basis for engendering wider polemic debate and signpost future academic endeavours.

## 2. Research Methodology

A mixed philosophical paradigm was adopted using both interpretivism [5,16–18] and post-positivism [19–21] for theory and model development, as this research considers existing theories subjectively while utilising an objectivist epistemological perspective [22]. This mixed philosophical approach has been widely used in the literature [23–26], and therefore, its use is justified in the present research setting. This is because this research seeks to understand and interpret safety risk models and theories subjectively from different viewpoints while also pursuing objectivity through post-positivism to minimise the possible impact of the researchers' bias'. Inductive reasoning, couched within a grounded theory strategy, was employed to answer the research question: can a novel theory and model for highway construction safety be developed and rationalised from existing models and theories of safety risk? Interpretivism adheres to a post-foundational epistemology with a purpose of interpreting and establishing facts within the context of the subject and experience being researched [17]. Post-positivism assumes that an approximation of phenomena under investigation could be objectively made, subject to individual researcher bias [27].

This qualitative research is conducted in three phases; see Figure 1. In phase one, the Preferred Reporting Items for Systematic Reviews and Meta-Analyses (PRISMA) was used in the search strategy to obtain bibliometric data on the diverse application of risk

models and theories to systematically review the extant literature and combine all relevant knowledge in this subject area [28]; each article constituted secondary data and a unit of analysis [29]. The Scopus journal database was adopted using keywords such as occupational and health or safety risk and accident theories or models. Only relevant subject areas and papers written in English were considered for this study. In phase two, VOSViewer was used to present data extracted from the literature search to facilitate a bibliometric analysis. A network map of co-occurrence of keywords and categorisation of significant keywords extracted was presented. Interpretivism was used to synthesise literature and critically analyse existing risk theories and models while employing the classification and tabularisation technique in phase three. An Excel spreadsheet was then used for manual classification analysis using color-coding of publications into thematic groups to assign publication into arbitrary thematic groups based on similar content within each paper. Current and future trends were then identified and considered in building a new conceptual model using the grounded theory approach.

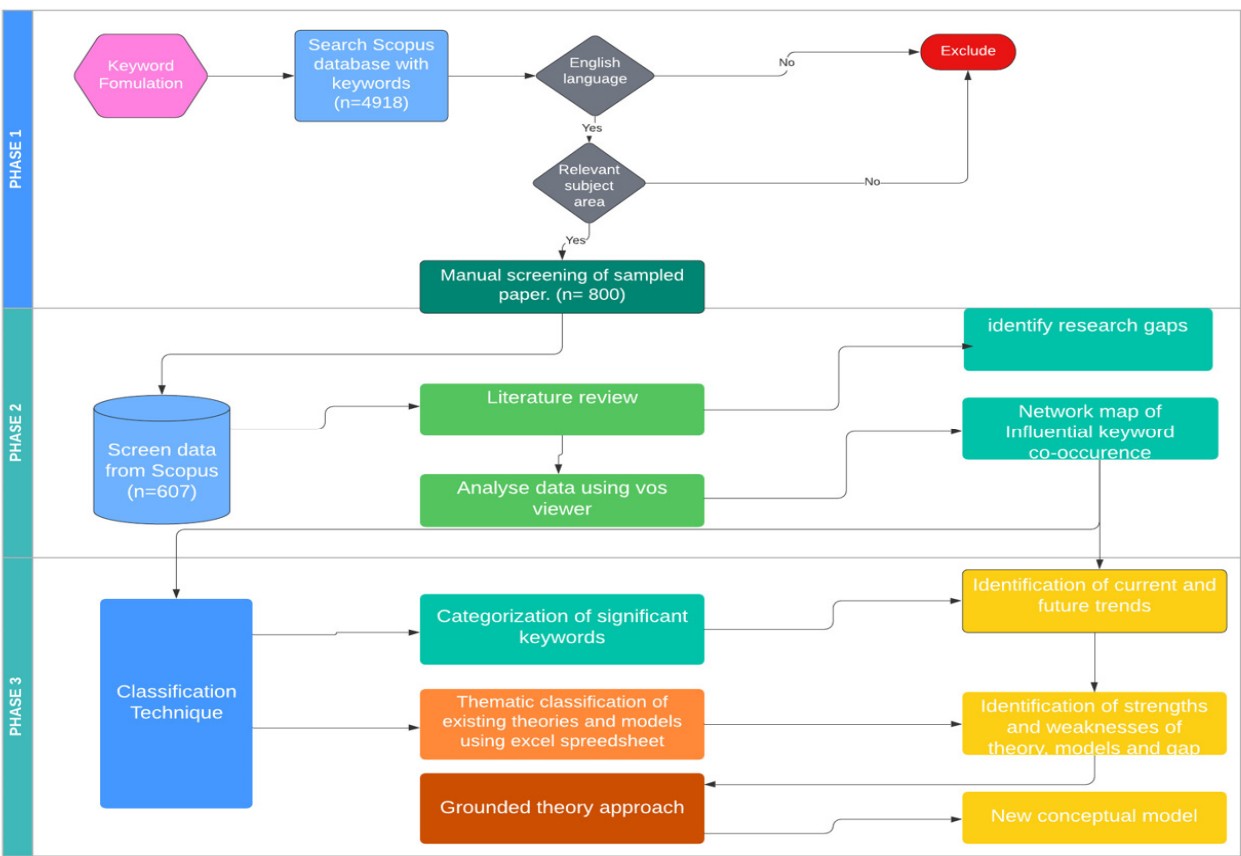

**Figure 1.** Methodology process diagram.

## 3. Bibliometric Search Technique

The articles selected for this review were published between 1990 and 2022. The year 1990 is selected as the starting point because the articles sourced from previous years were observed to have limited impact on the research. Literature review papers were retrieved from the Scopus database, as it covers almost all journals and publications when compared to alternative digital repositories [30–32]. To search for journal articles, the title and keyword search rule used was 'TITLE-ABS-KEY (occupational AND health OR safety OR risk AND accident AND theories OR models). This resulted in sampling 4918 documents in the preliminary search. This was refined to reduce the number of articles by selecting journals that are relevant to the study domain. This included engineering, safety, decision sciences, business management, computer science, social science and environmental science domains.

Some areas, such as finance, economics, chemistry, medicine and nursing, were excluded from the search as these were irrelevant to the study. Consequently, 800 articles were extracted from the database. The papers selected were then manually screened, and superfluous papers removed, leaving a final data set of 607 papers that covered various industrial sectors (e.g., oil and gas, aviation, mining, medicine and building construction).

## 4. Classification Technique

Using the bibliometric data retrieved from Scopus, a thematic analysis was performed to identify the various categories of models and theories presented by individual papers. Models and theories presented by the papers were analysed to understand their operational patterns, which were classified into groups. Based on the purposes, applications, features and domain of the theories and models sampled, they were classified into seven different groups: (1) element models/theories; (2) incentive models/theories; (3) quantitative and statistical models/theories; (4) behavioural models/theories; (5) sequential models/theories; (6) barricade models/theories; and (7) resilient models/theories. Similarly, papers based on generic risk management and safety management systems were considered for further understanding of how these theories and models have been applied across the various fields. The classification strategy is a technique of analysing data by discovering common ideas that run through different literature and grouping those with similar ideas into a common group [33,34].

## 5. Research Results

Using VOSViewer, analysis of all keywords present in the selected literature was conducted by selecting keywords that occurred at least five times. Of all keywords (i.e., 2593), only 162 keywords met this threshold. 'All keywords' was used as a criterion in keyword selection instead of 'author keywords' or 'index keywords' to present a more detailed and comprehensive picture of the prevailing academic discourse on the phenomena under investigation. Selecting 'all keywords' prevented bias when viewing topics within the subject based on the authors' perspective and knowledge. However, it was recognised that this could result in superfluous information within the visualization, making it complex and difficult to interpret or manipulate [35]. Therefore, keywords that had little impact in terms of weighting, synonymous terms (such as 'human' or 'humans'), and uninfluential keywords such as male, female, article, priority journal, etc., were manually screened out, resulting in sampling 141 keywords for analysis.

## 6. Co-Occurrence of Keywords

The network visualisation (Figure 2) uses an overlay illustrating eleven prominent keywords that indicate topical areas where risk theories and models have been applied. These include 'occupational risk' (frequency ($f$) = 121); 'accident prevention'($f$ = 108); 'safety' ($f$ = 63); 'human' ($f$ = 82); 'risk assessment' ($f$ = 64); 'accident' ($f$ = 63); 'occupational accident' ($f$ = 56); 'occupational safety' ($f$ = 55); 'risk management' ($f$ = 31); 'safety management' ($f$ = 42); and, from a wider perspective, the 'construction industry' ($f$ = 43). The predominance of these keywords highlights the adoption of theories and models for enhancing occupation health and safety to prevent accidents. Furthermore, the green and yellow color-coded nodes illustrate that the keyword 'human' ($f$ = 82) is predominantly used between 2013 and 2014, and 'construction worker' ($f$ = 13) and 'worker' ($f$ = 7) are more recently adopted, between 2018 and 2021, and have the same meaning. This could imply that the focus on the human element in theory and model building remains prominent in literature. The keywords in yellow show the most recent trends and directions of models with a safety focus on 'prediction' ($f$ = 7); 'project management' ($f$ = 9); 'human resource management' ($f$ = 23), 'construction worker' ($f$ = 13); 'occupational injury' ($f$ = 7); 'construction equipment' ($f$ = 5); 'procedures' ($f$ = 5); and 'risk perception' ($f$ = 9). This trend suggests a shift from reactive systems such as 'laws and legislation' ($f$ = 8) and 'risk

reduction' ($f = 9$) between 2010 and 2012 to more proactive systems such as 'prediction' ($f = 7$) and human resource management during 2016–2018.

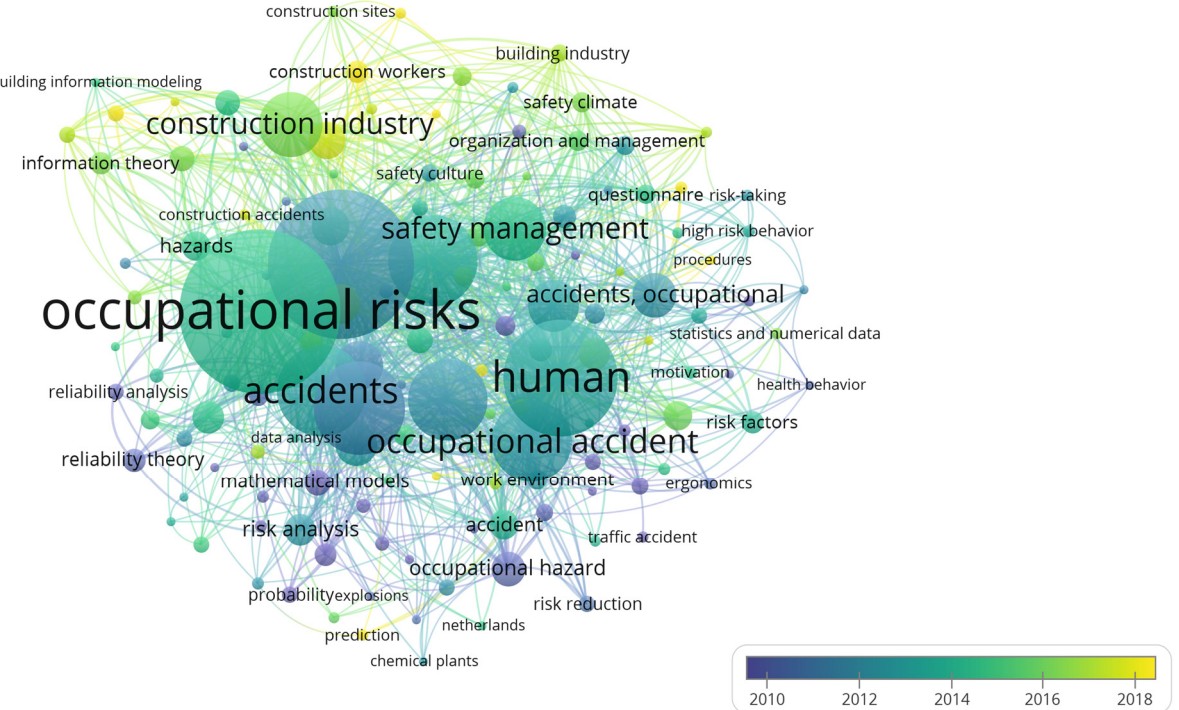

**Figure 2.** Network map of keyword co-occurrence.

## 7. Existing Safety Risk Theories and Models

The purple, blue and deep green nodes show a trend of keywords that dominated between 2010 and 2015. This trend displayed keywords such as 'risk analysis' ($f = 12$), 'occupational hazard' ($f = 20$), 'occupational risk' ($f = 121$), 'occupational accident' ($f = 56$), 'human' ($f = 82$) and 'safety management' ($f = 42$). These keywords demonstrate the focus of existing theories and models of safety risk. In comprehending and significantly influencing safety risk challenges, theories and models have been developed to effectively explore occupational safety risk problems in terms of occupational accidents and occupational hazards. For example, Lees [36] justified the effectiveness of models and theories in the investigation of safety risk incidents and accidents. Investigating an incident is essential to understand why it occurred in order to forestall future occurrence. Early theories and models have seen several researchers propose theories and techniques to aid this process [11,37–40]. For example, Heinrich [14] proposed a domino theory, where the central premise focused on human behaviour. A modification to Heinrich's theory [14] was proposed by Bird et al. [41] by including management and organisational aspects in the causal factors in incidents. Whittington et al. [42] discovered the role of management errors as major contributors to incidents in the construction industry. On the issue of worker distractions, Hinze [43] proposed a distraction theory, where the risk of a construction incident increased due to worker distractions.

Abdelhamid and Everett [44] developed a model for identifying root causes of construction incidents, such as worker attitudes, training and procedures. The limitation with this model, however, is attributed to the fact that it ignores systemic root causes that might explain why a procedure, for example, failed and caused an incident. Following this limitation, Gibb et al. [45] proposed additional inquiries targeted at leadership, culture, project management decisions and design inadequacies. Proximal and distal factors in construction incidents were identified and proposed by Suraji et al. [46] as an incident causality method. While factors directly related to the incident cause were termed as proximal factors, the

research [46] defined distal factors as those that lead to the introduction of proximal factors, such as design complexity and time constraint.

According to Reason [47], contradiction incident causation theories can be encapsulated by the 'systems approach' and 'person approach'. The person approach centres on the errors of individuals, such as inattention, forgetfulness, or moral weakness, whereas the system approach focuses on conditions under which the individual works and attempts to build defences to avert errors or mitigate their effects. A comprehensive management program aimed at several different targets, such as the person, task, team, workplace and the institution as a whole, is needed to make the systems approach work effectively [48]. According to Burgoyne [49], maximum attention to detail must be placed on investigating an incident, with the aim of extracting the highest amount of knowledge from the experience and disseminating the knowledge in a way that forestalls future occurrences and ensures personnel safety. The term 'root cause' is widely used in practice and must be explored to the point where there is nothing further to investigate. Investigators often find difficulty in reporting deep-rooted findings of incidents that are as a result of organisational policy or culture, and hence mostly settle on employee-based causes [50].

## 8. The Trend and Future Direction of Theory and Model Building

The keywords in yellow show the most recent trends and direction of models in safety risk focus on 'project management' ($f = 9$); 'human resource management' ($f = 23$); 'construction worker' ($f = 13$); 'occupational injury' ($f = 7$); 'construction equipment' ($f = 5$); 'procedures' ($f = 5$); 'risk perception' ($f = 9$); and 'prediction' ($f = 7$). Recently, the fragmented nature of construction work activities has made it imperative that safety risk is considered on a project basis instead of the previously generalised sole safety considerations made for the entire organisation [51,52]. Safety risk is now managed on a project basis as a divide and conquer approach [53], highlighting the influence of project management on safety risk management [8,54]. The models and theories reviewed predominantly focus on human error as a major cause of accidents, and it is not coincidental that 'construction worker' and 'human resource management' feature as trends within model building. Human resource management is a vital contribution to the developing domain of safety culture, which requires a modification in staff safety perception and work behaviour to prevent occupational injury [55]. The procedure involved in task performance and the equipment used is another trend. Although human errors are considered a root cause of accidents, in some instances, human errors are only a reflection of system lapses.

Any ambiguity in the description of work procedure [37], inadequate personal protective equipment or faulty machines could cause human errors that result in accidents or injury, hence the requirement to investigate these factors and include them within theory and model building. One major change of safety risk management in current trends is a notable shift from reactive systems such as 'laws and legislation' ($f = 8$) and 'risk reduction' ($f = 9$) between 2010 and 2012 to more proactive systems such as 'prediction' ($f = 7$). Thus, prediction of safety risks and accidents before they occur (via lead indicators) as a preventive method has become more important than mitigation methods, which are more reactive.

## 9. Categorisation of Significant Keywords

The clustering of the keywords displayed in the map projected certain keywords that highlighted a number of perspectives on the trend of theory/model adoption and application. These conspicuous keywords were grouped and interpreted in Table 1 as the prominent theories most considered; the domains in which these theories/models have been applied most; the key safety indicators presented within the map; the data collection methods applied most frequently. Another noticeable element within the map was the various keywords used to describe workers. This could stem from how significant human elements are considered in safety model building. The highway construction industry is also not featured in the domain area for model application. This could support

the proposition that scant research has been done on highway worker safety, at least in academia. Instead, the analysed papers showed that the health and safety of workers has been generalised to encompass all construction activities, and consequently the term 'construction worker' is the generalised term used to describe the staff in the construction industry, with no mention of highway workers specifically.

**Table 1.** Keyword categorization.

| Category | Keywords | Frequency (f) | Percentages (%) | Citations |
|---|---|---|---|---|
| *Prominent Theories/models* | Information theory | 13 | 8.8 | [56,57] |
| | System theory | 25 | 17.0 | [37,58] |
| | Theory of planned behaviour | 11 | 7.5 | [14,59] |
| | Reliability theory | 14 | 9.5 | [60,61] |
| | Mathematical models | 16 | 10.9 | [62] |
| | Statistical models | 5 | 3.4 | [63] |
| | Bayes theory | 5 | 3.4 | [64,65] |
| | Bayesian network model | 9 | 6.1 | [66,67] |
| | Fuzzy set theory | 9 | 6.1 | [10,68] |
| | Structural equation model | 11 | 7.5 | [69] |
| | Regression analysis | 8 | 5.4 | [58] |
| | Decision theory | 13 | 8.8 | [70] |
| | Laws and legislations | 8 | 5.4 | [39,71] |
| *Domain* | Construction industry | 43 | 36.4 | [4,72] |
| | Building information modelling | 14 | 11.9 | [20] |
| | Tunnel construction | 11 | 9.3 | [73] |
| | Coal mines | 7 | 5.9 | [13] |
| | Traffic accidents | 6 | 5.0 | [38] |
| | Transportation | 5 | 4.2 | [74] |
| | Building industry | 10 | 8.4 | [18,20] |
| | Mining | 7 | 5.9 | [75] |
| | Agriculture | 6 | 5.0 | |
| *Definition words for human factor* | Operatives | 8 | 24.2 | [76] |
| | Worker | 7 | 21.2 | [77] |
| | Construction worker | 13 | 39.4 | [1] |
| | Employee | 5 | 15.2 | [12] |
| *Key Safety-Indicators (factor)* | Safety climate | 12 | 12.6 | [78] |
| | Safety culture | 11 | 11.5 | [68,73] |
| | High-risk behaviour | 7 | 7.4 | [76] |
| | Risk perception | 25 | 26.3 | [79] |
| | Safety behaviour | 10 | 10.5 | [12] |
| | Task performance | 5 | 5.3 | [37,48] |
| | Industrial hygiene | 20 | 21.1 | |
| | Construction equipment | 5 | 5.3 | |

## 10. Information Theory, Digital Technology and Safety Models

Information theory is an emerging area in the health and safety domain, with an average publication year of 2016 and $f = 13$ occurrences in the keyword analysis; it represents the study of storing, measuring and transmitting digital information [80]. This is of particular interest because it encapsulates all the prominent theories and models, including the statistical models, Bayes theory, Bayesian models, etc., listed in Table 1. The type of information that this theory handles is uncertainty-based; hence, it is considered a useful component in influencing highway construction safety, which is characterised by fuzziness and uncertainty. Information theory has recently become prominent in the occupational health and safety domain. For example, Zhuang et al. [81] implemented information theory in investigating safety passage planning for system shoring supports with building information modelling (BIM). Bauk et al. [57] adopted it for communication to increase occupational safety at a seaport, enabling a powerful marine and offshore decision-support

solution through a Bayesian network technique [66]. The impact of information theory has been crucial to its application in areas such as model selection [82], data analysis [56], pattern recognition, and anomaly detection [83]. Digital information technology advancements such as BIM have been influential in safety management [84] and have required the digitisation and digitalisation of safety-related data. The close relationship between ontology and information technologies can be explored to develop knowledge-based safety models [85,86], which could analyse and predict safety risks and accidents [58]. Digital data analytics could also be adopted in implementing leading safety indicators, useful in safety risk prediction [87].

## 11. The Relationship between the Human Element, Key Safety Indicators and Domain Application

Theories and models have focused on human-centred themes to continually investigate and provide measures for errors [76,77]. This is evident in the various keywords used in emphasizing human involvement in the safety risk literature, such as operatives, workers, etc. However, humans remain versatile and constantly alter their actions based on situations rather than procedures [88]. These actions could be influenced by key safety indicators, such as worker perception of risk, behaviour, safety climate, and safety culture [68,79,89]. Therefore, a focus on human errors has been expanded to include more proactive indicators, such as risk perception, safety climate, and safety culture, in contemporary safety risk models and theories to address the human contribution to safety risk events. The different ways workers perceive risk, their performance and their attitudes create behavioural patterns, which can be associated with their demography [90] and safety climate factors like co-worker attitude and safety awareness [76]. Other studies reveal the integral part safety climate plays in safety culture [91,92]. Safety climate has a situational, behavioural and psychological influence on safety culture [79]; hence, analysing safety climate could enable the prediction of safety culture and further impact safety performance. Human resource management is also an emerging trend between 2018 and 2021. This is particularly important given that highway construction is characterised by different groups of contract workers and contingent workers, such as highway engineers, road maintenance operatives, traffic officers and inspectors.

## 12. Predictive Safety Models and Big Data

Predictive models provide insight into future events [93]. From the scientometric analysis and categorisation of the keywords, it is inferred that safety risk management is moving from a reactive approach to a more proactive approach through prediction for prevention. The trend in using prediction in safety management provides a proactive approach to handling risks and accidents. Predictive models and frameworks are associated with planning and strategy, with specific emphasis on foreseeing future disruptions in a system [58,87]. Safety risk data in construction is characterised by its large volume, velocity, value and variety, which are all features of big data [94]. Therefore, big data analytics can be used in building machine learning predictive models based on existing and historical data available with a more robust storage facility. Gerassis et al. [67] used data mining and attribute selection techniques to identify the main predictors of accidents related to embankment construction, then built Bayesian networks to predict the individual causes of various accident types. Amiri et al. [65] employed decision trees, association rule and multiple-correspondence analysis, which resulted in identifying a significant relationship between time of accident, accident location and the part of the body impacted by the accident. Workers who are mostly prone to fatal accidents have also been identified and classified using Bayesian theory; by ranking the risks involved, the result was applied to training workers to reduce accidents and safety risk issues on site [64]. Knowledge from these previous works could be employed in building an accident prediction model for highway workers.

## 13. Classification of Models

From the classification analysis technique and grounded theory analysis, the reviewed models and theories were categorised into seven thematic classifications based on their features, processes, domain, idea and application; each category's benefits and limitations were also compared to reveal the strengths and weaknesses of these classified theories and models (see Table 2). The dearth of studies reviewed demonstrates the benefits and limitations of each of the seven model categories presented.

**Table 2.** Model classification.

| Model/Theories | Features | Benefits | Limitations | Authors |
|---|---|---|---|---|
| Element Theories/Models | Characterizes the unique participating elements and components that make the model | 1. Its holistic nature allows individual weakness and solutions to be identified<br>2. Reveals individual parts' contribution to model<br>3. Enables proper distribution of human resources and structures | 1. Does not identify the interconnection between components<br>2. May not reveal political or representational elements<br>3. Analysing actions and processes are challenging<br>4. Results are difficult to describe | [11,37–40,95–97] |
| Incentive Theories/Models | Characterizes the actions and initiatives that enhance safety with feedback elements | 1. Identifies the role management can play to enhance safety<br>2. Highlights contributions and responsibilities of workers and management<br>3. Enables proper distribution of human resources and structures<br>Robust insight into results | 1. Does not identify the interconnection between components<br>2. May not reveal political or representational elements<br>3. Its focus on specificity may make it unable to be applied holistically | [39,77,98] |
| Quantitative or Statistical | Identifies the connection between events and incidents by quantifying data and finding patterns | 1. It has evidence-based outcomes<br>2. Decisions taken are backed by evidence<br>Gives insight into results and relationships | 1. Its focus on specificity may make it unable to be applied holistically<br>2. Does not recognize all components<br>Contextual considerations need to be made when in use | [62,63,99] |
| Sequential | Identifies a chain of events that leads to an accident | 1. Gives insight into causes of failure<br>2. At the micro level, provides good examination<br>3. Relationships between causal components are described explicitly<br>Focuses on human error and how it could be prevented | 1. Does not reveal contributions to success<br>2. At a strategic level, it is challenging to apply to more complex systems<br>3. Effects are not identified<br>May not reveal political or representational elements | [14] |
| Behavioural | Identifies the unsafe behaviour and attitude of workers as leading causes of accidents | 1. Normally based on safety climate<br>2. Could be generally applied to several situations<br>3. General focus on parts and purpose<br>Reveals human resource management qualities | 1. Its generalization feature may prevent it from being used in specific instances<br>2. May not reflect on all connections<br>3. Challenging to recognize all chances for solutions<br>Guidance on specific activities and situations is scarce | [69] |

**Table 2.** *Cont.*

| Model/Theories | Features | Benefits | Limitations | Authors |
|---|---|---|---|---|
| Barricade | Safety is measured based on the effectiveness of barriers erected | 1. Provides defence against failure 2. Gives insight to causes of failure 3. At the micro level, provides good examination 4. Relationships between causal components are described explicitly | 1. Does not reveal contributions to success 2. At a strategic level, it is challenging to apply to more complex systems 3. Effects are not identified | [100,101] |
| Resilience Models | Focuses on the ability of a system to handle varying conditions and how long it takes to restore to normal conditions after a disturbance occurs | 1. Gives insight into contributing successes 2. Provides shocks for the system 3. Allows system to return to normal function after failure | 1. Cannot predict future situations 2. Only focuses on present  The magnitude of relationships and components could make it too complex for application | [102–105] |

All models and theories identified in this review could potentially be applied to highway worker safety risk mitigation; however, many components characterise highway construction, and although these components work comparatively independently, they interact closely with each other over a period. Therefore, individual purposes and context determine which model or theory will be applicable. A comparison of model features based on resilience theory [102,105] with the models classified in Table 2 showed that the classified models are not entirely independent of each other and demonstrate some common features. While these different models propose different means for safety risk mitigation, only models based on resilience theory provide alternatives for adaptability and recovery, even if incidents or accidents occur. The resilience theory approach has the ability to inculcate all significant factors, elements and results of the other models classified when thoroughly applied to highway construction.

## 14. Resilient Predictive Models

Resilience involves the inherent capabilities of a system to adapt its operation before or after any modifications and disruptions to carry on with performing its functions when hit by a major catastrophe [102]. This is based on resilience theory [105]. Resilience promotes a holistic view of how a system adjusts dynamically and changes to promote the continuity of safer operations [103]. It focuses on effectively using resources to proactively anticipate and manage risks [79]. This, however, is not the same as predictive modelling, although there are similar characteristics [106]. While predictive models handle future disruptions and focus on planning for these, resilience engineering (resilient models) focuses on the current system and its response to disturbances, its adaptability and the pace for recovery back to normal activities. The resilience measures (data) generally recognised include awareness, safety culture, anticipation, management commitment, flexibility, reporting culture and safety climate [105,107,108]. Adaptability, response to incidents and system recovery are gaps in predictive models that could be addressed by introducing resilient features into the predictive models. This study therefore proposes a digital predictive model that has resilience capabilities. This digital model will predict the probability of safety risk events occurring, and if they are unavoidable, resilience features will provide shocks to enable recovery.

## 15. Discussion

Early safety risk theories and models [14,42] focused on human errors and behaviour as the major risk contributors in an organization. Others, e.g., [41], later included lack of or inadequate management responsibilities in their models. Several other safety risk models were developed based on earlier proposed models, and irrespective of improvements made, the human factor in safety risk management was not made redundant. This demonstrates how indispensable humans are in safety risk model/theory building; any model developed to address safety risk challenges should consider human factor elements and include key leading indicators that proactively measure human involvement in safety risk events.

The keyword co-occurrence analysis showed that the focus of safety risk research is moving towards 'project management', 'human resource management', 'construction worker', 'risk perception' and 'prediction'. The literature reveals that safety risk challenges are being handled on a project basis to personalise safety risk measures and put solutions in place. The keywords 'human resource management' and 'construction worker' highlight the concentration on the human factors in safety risk, and this is emphasised by the different terms used to describe humans in the categorised keywords section. Theories and models have focused on human-centred themes to continually investigate and provide measures for errors. However, the versatility of humans remains constant; they respond to situations rather than procedures, and this has led to varying human safety indicators to look beyond error to including leading indicators such as risk perception, safety climate, safety culture and worker behaviour (high-risk behaviour). The keyword 'prediction' was also presented as a recent trend and future direction for safety risk models and theory. Existing models and theories focus more on a reactive mitigation of risk after it has occurred rather than proactive prevention of risk before it occurs [109]. Predicting safety risk could help avert misfortune and accidents.

Inculcating information technology into safety risk model research and digitalization is apparent in current trends [110]. Keywords such as 'information theory', 'Bayes theory', 'Bayesian network model', 'fuzzy set theory' and 'regression analysis' indicate the presence of technology in the field of machine learning and data analytics in safety risk model research. Information theory and fuzzy sets have been widely used to identify patterns in data for analytical insight [111], while Bayesian network models and regression analysis models have predictive abilities that can be applied to predict safety risk levels before they occur.

Based on similar features, processes, domains, ideas and applications, the various models identified were classified. Comparing the strengths and limitations of the models and theories revealed that the variety of models classified are not entirely exclusive and demonstrate features that are inherent in other models. However, individual purposes and context determine which model or theory will be applicable. From the literature, it was discovered that only models based on resilience theory (resilient models) had the ability to adapt and recover after an incident occurs. Predicting safety risk incidents or safety risk levels is a huge step toward the prevention of an accident or loss occurrence in an organisation. Nevertheless, predicting future occurrences only helps in anticipating and planning for these risks, without any provision for adaptability and recovery in the event that the risks predicted are unavoidable. This leaves a gap for digital predictive models that have inherent resilient features capable of recovery in the event of an inevitable predicted safety risk.

Premised upon the analysis of literature, this paper proposes a concept of a digital predictive model enhanced with inherent resilient features (data) to develop a conceptual framework known as a resilient predictive model ($P_r$) (see Figure 3).

The resilience measures (data) generally recognised from the literature include awareness, safety culture, anticipation, management commitment, flexibility, reporting culture and safety climate. These data are collected through secondary data such as worker satisfaction surveys and accident data as well as primary data collection through interviews with highway workers. These data are added to variables that will be incorporated in building

a digital predictive machine learning model. The types of models studied have provided diverse knowledge in the field of safety risk mitigation. While some have taken a holistic approach towards mitigation and control of construction risk, others have dealt specifically with other domains, such as mining, coal mining, tunnel construction, etc. However, no presently available model has adopted a predictive model fused with resilience capabilities in the highway construction domain.

The data collected are pre-processed and cleaned through data transformation and reduction techniques such as normalization and attribute selection through goodness of fit chi-square test to identify whether there is an association between predictive and resilient variables in the dataset and whether the sample represents the whole population. This process decreases the size of the data, limiting it to only the most important information and increasing the accuracy and efficiency of machine learning models. Machine learning models such as Bayesian models, support vector machines and random forest, as well as deep learning models such as artificial neural networks and recurrent neural networks are applied to the data to build different predictive classification models. The model building process involves a random selection of 80% of the data for training and 20% for testing and verification.

The different models built are compared, evaluated and validated. Metric that are used to evaluate the machine learning and deep learning algorithms include Classification Accuracy, Logarithmic Loss, Confusion Matrix, F1 Score, Mean Absolute Error and Mean Squared Error. K-fold cross validation and bootstrapping are then employed in validating the results of the model to properly understand the model and estimate an unbiased generalization performance.

By presenting this idea for a digital model now, it is anticipated that the work will engender wider polemic debate in this area of investigation that will not only refine the proposed model, but will also contribute towards a safer highway construction environment. Future work will test the model in practice to ensure that predictions are accurate.

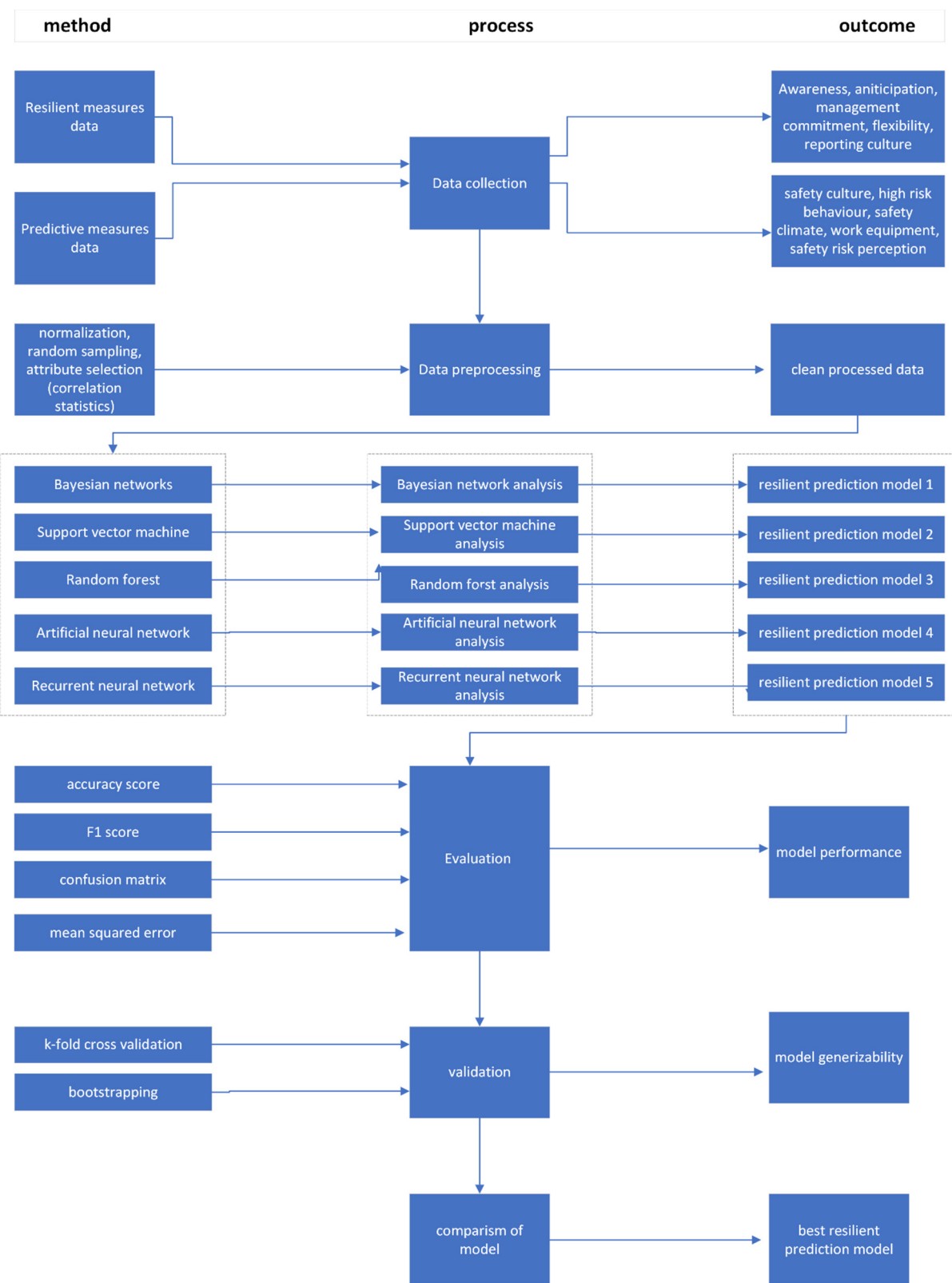

**Figure 3.** A schematic diagram of the resilient predictive conceptual model.

## 16. Conclusions

Safety and risk are omnipresent challenges confronting the construction industry, despite the health and safety improvements that have been made. Previous research has provided numerous insights into fostering construction safety performance, including the

construction of models and theories that guide safety activities. This paper adopted a bibliometric mapping approach by conducting a comprehensive keyword analysis to identify new trends in model building using VOSViewer software and based on 607 papers extracted from Scopus [112,113]. The methods employed included keyword co-occurrence analysis and classification techniques, where models and theories were categorised based on similar themes and ideas. Based on the screening of topics and abstracts, areas of active research were identified, namely 'prediction', 'project management', 'human resource management', 'construction worker', 'occupational injury', 'construction equipment', 'procedures' and 'risk perception', with prediction and human resource management being prominently investigated during 2018 and 2021. Further analysis of keywords through categorisation revealed five prominent clusters, which included 'prominent theories/models', 'domains' (where models/theories have been applied), 'definition of worker' (terms used to describe worker), 'key safety indicators' and 'data collection methods' employed. This study identified research gaps, which included (1) the information theory and technology gap has not been well exploited in building theories and models for safety; (2) the under-exploration of theories and models for highway construction, as these have been largely generalised to encompass all construction industry activities, hence leaving highway workers vulnerable; and (3) the need for a resilient prediction model that inculcates leading safety indicators and technology such as machine learning (pattern recognition and prediction), data analytics and the storage method for digitised data. The scientometric keyword analysis provided insight for future research directions, such as applying big data technologies (data analytics, machine learning, data storage) in safety management.

'Key safety indicator' was another influential category and included keywords such as 'safety climate', 'safety culture' and 'risk perception', which suggests emphasis on moving towards leading safety indicators as proactive measurements. The emerging trend of leading indicator data such as 'safety climate', 'safety culture' and 'risk perception' and reactive safety indicator data such as 'high-risk behaviour', 'safety behaviour' and 'task performance' were inculcated into building a resilient model on the basis of resilience theory, while big data technologies such as data analytics and pattern recognition were inculcated into building a predictive model based on information theory principles. These two concepts were combined to create a resilient predictive safety model, which is proposed in this paper as an alternative method for addressing highway construction risks. This will help management in making safer decisions based on empirical knowledge.

This work is in its early conceptual stages. This concept does not evaluate the most significant factors of resilience and prediction that have a major impact on highway construction safety. This is a conceptual model under development. Future works will build a prototype model by exploring in-depth the proposed model by investigating various machine learning architectures that could be employed in building predictive models with resilient model data, using deductive and deterministic modelling to test the new model. In addition, user adaptability and acceptance as well as industry readiness to employ various big data technologies in construction safety management will be assessed. These proposed future directions could help both industry practitioners and academics in safety performance enhancement to improve worker health, safety and wellbeing. It is worthy to note that all of the selected literature was sourced from Scopus and in English only, which could potentially exclude some relevant studies published in other languages or on different platforms.

**Author Contributions:** Conceptualization, L.B., D.J.E., C.R. and I.R.; methodology, L.B., D.J.E. and C.R.; validation, L.B. and D.J.E.; formal analysis, L.B., D.J.E., C.R. and I.R.; investigation, L.B. and D.J.E.; data curation, L.B. and D.J.E.; writing—original draft preparation, L.B., D.J.E. and C.R.; writing—review and editing, L.B., D.J.E., C.R. and I.R.; supervision, D.J.E., C.R. and I.R.; project administration, L.B., D.J.E., C.R. and I.R. All authors have read and agreed to the published version of the manuscript.

**Funding:** This research is supported by National Highways.

**Institutional Review Board Statement:** The study was conducted according to an ethical protocol that was approved by the Computing, Engineering and the Built Environment Faculty Academic Ethics Committee of Birmingham City University (Edwards/#7741/sub1/Mod/2020/Sep/CEBE FAEC-BNV6200 ACM Version D.J. Edwards-13 October 2020).

**Informed Consent Statement:** Informed consent was obtained from all subjects involved in the study.

**Data Availability Statement:** Anonymized data are available from the corresponding author upon written request and subject to review.

**Conflicts of Interest:** The authors declare no conflict of interest.

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
