# Peer review of "A Review of Safety Risk Theories and Models and the Development of a Digital Highway Construction Safety Risk Model"

_digital, doi:10.3390/digital2020013_

Round 1
Reviewer 1 Report
The paper gives a broad overview of the relevant literature. No other types of research, other than a literature review, have been performed. So the 'resilient predictive conceptual model' is only based on this review and therefore purely theoretical. How sure can you be that this model will work?
Author Response
Thank you for your comments and suggestions - we do appreciate your input. We concur that the paper focuses on using literature to develop a new theoretical model. However, the authors have stated that the theoretical model developed will require future testing in practice to ensure a high degree of prediction accuracy. A new sentence in the discussion section has been added to make this point more explicit. Once again, thank you for your comments made.
Reviewer 2 Report
The authors processed the information only on the basis of keywords, which articles dealt with this issue and at the same time qualitatively described the outputs. If there is a possibility of publishing only a search article review", then this is fine.
However, the authors, resp. author D.J.Edwards probably (as far as the same person is concerned, but the name is identical D.J.Edwards) quotes in this article about 20 times, I don ´t know if it is appropriate. In the case of the possibility of publishing a review article, I recommend modifying the used literature, excluding self-citations.
Author Response
Yes, this is correct, we used keywords to conduct a search of literature. This is standard practice in academia (you must identify keywords before conducting a keyword search of journal databases) and we thank you for the comment and observation.
Round 2
Reviewer 2 Report
In my opinion, the article contains a significant number of self-citations.